# In Silico Study of Novel Cyclodextrin Inclusion Complexes of Polycaprolactone and Its Correlation with Skin Regeneration

**DOI:** 10.3390/ijms24108932

**Published:** 2023-05-18

**Authors:** René Gerardo Escobedo-González, Edgar Daniel Moyers-Montoya, Carlos Alberto Martínez-Pérez, Perla Elvia García-Casillas, René Miranda-Ruvalcaba, María Inés Nicolás Nicolás-Vázquez

**Affiliations:** 1Department of Industrial Maintenance, Technological University of the City of Juárez, Av. Universidad Tecnológica No. 3051, Col. Lote Bravo II, Ciudad Juárez 32695, Mexico; 2Institute of Engineering and Technology, Autonomous University of the City of Juárez (UACJ), Ave. Del Charro 450 Norte, Ciudad Juárez 32310, Mexico; edgar.moyers@gmail.com (E.D.M.-M.); camartin@uacj.mx (C.A.M.-P.); perla.garcia@ciqa.edu.mx (P.E.G.-C.); 3Applied Chemistry Research Center, Blvd. Enrique Reyna Hermosillo No. 140, Saltillo 25294, Mexico; 4Chemical Sciences Department, UNAM–FESC, Campus 1, Cuautitlán Izcalli 54740, Mexico

**Keywords:** inclusion complexes, β–cyclodextrin, polycaprolactone, in silico study

## Abstract

Three novel biomaterials obtained via inclusion complexes of β–cyclodextrin, 6-deoxi-6-amino-β–cyclodextrin and epithelial growth factor grafted to 6-deoxi-6-amino-β–cyclodextrin with polycaprolactone. Furthermore, some physicochemical, toxicological and absorption properties were predicted using bioinformatics tools. The electronic, geometrical and spectroscopical calculated properties agree with the properties obtained via experimental methods, explaining the behaviors observed in each case. The interaction energy was obtained, and its values were −60.6, −20.9 and −17.1 kcal/mol for β–cyclodextrin/polycaprolactone followed by the 6-amino-β–cyclodextrin-polycaprolactone complex and finally the complex of epithelial growth factor anchored to 6-deoxy-6-amino–β–cyclodextrin/polycaprolactone. Additionally, the dipolar moments were calculated, achieving values of 3.2688, 5.9249 and 5.0998 Debye, respectively, and in addition the experimental wettability behavior of the studied materials has also been explained. It is important to note that the toxicological predictions suggested no mutagenic, tumorigenic or reproductive effects; moreover, an anti-inflammatory effect has been shown. Finally, the improvement in the cicatricial effect of the novel materials has been conveniently explained by comparing the poly-caprolactone data obtained in the experimental assessments.

## 1. Introduction

Nanomaterials in recent years have been the subject of numerous studies due to their interesting properties, such as optical, magnetic, electronic and acoustic properties [1,2,3,4,5]. Nanomaterials, particularly those of the polymeric type, have been therapeutically used to control inflammation [6], in drug delivery systems [7,8] and for their scaffolding function [9].

Polycaprolactone (PCL) is a semi-crystalline aliphatic polyester, with an easy-to-handle shape, and is a biocompatible molecule with wide biomedical applications. This interesting material has been approved by the American Food and Drug Administration (FDA) for its application in biomaterials and biomedical devices. However, its biological applications are limited by the absence of an active site and its high hydrophobicity [10,11,12,13]. Consequently, it is very important to stimulate the modification of PCL materials to improve the cell–material interaction [14,15,16,17].

In recent works, one approach to add bioactive sites to PCL and other hydrophobic polymers has been the preparation of the non-covalent inclusion complex (IC) with cyclodextrins (CDs) [18,19,20], which are cyclic molecules comprising 6, 7 or 8 glucose units (α, β or γ) with an internal cavity with high hydrophobicity [21,22,23,24,25]. In this sense, the pseudo polyrotaxanes and polyrotaxanes are materials obtained via the complexation of linear polymeric chains with cyclodextrin (α– and γ– generally); it is important to comment that this class of molecules is useful for biomedical applications [19,26,27].

Recently, computational chemical methods, such as quantum chemical calculations and cheminformatics tools, have been used in the study of geometrical, spectroscopic and biological properties of many molecules, highlighting their employment in designing new drugs [28,29,30,31,32,33,34,35,36,37]. It is important to mention that these theoretical methods alone or complemented with the obtained experimental results can offer advantages to the design, in addition to predicting and explaining the structural properties in miscellaneous nano- and biomaterials [38,39,40]. Additionally, the achieved theoretical results provide excellent support to appropriately assign and correlate both bands and signals from several analytical techniques: UV-VIS, Raman, infrared and NMR [41,42,43,44]. In this sense, density functional theory (DFT) calculations have been useful and competent in predicting materials’ properties [39,43,45,46].

It is important to highlight that our research group has reported the obtention and biological evaluation of three novel nanofibers obtained via the inclusion complexes of β–cyclodextrin and some of its derivatives with PCL [47,48]. The studied materials were β–cyclodextrin/PCL (β–CD/PCL), 6-deoxy-6-amino-β–cyclodextrin/PCL (β–CDNH_2_/PCL) and epithelial growth factor anchored to 6-deoxy-6-amino-β–cyclodextrin/PCL (EGF–β–CDNH_2_/PCL); see Figure 1. β–CDNH_2_/PCL and EGF–β–CDNH_2_/PCL showed the best cell viability and cicatricial effect, in comparison to PCL and β–CD/PCL.

Considering both the importance of these materials and the requirement of appropriated knowledge of the interactions with the polymeric matrix in biological uses [44,49], our research group deemed it convenient to report the novel and interesting results obtained via a theoretical study of the target materials of this work. Therefore, a quantum chemistry approach was used to calculate the minimum energy of the designed structures, also determining their geometrical parameters and spectrophotometric data using IR and Raman techniques. Additionally, the structure–activity relationship was determined by comparing the quantum electronic parameters with the biological test results, producing molecular predictions regarding their activity and employing cheminformatics methods.

## 2. Results and Discussion

In a previous piece of work, we reported three novel biomaterials [47,48] which were designed to exhibit properties that emulate the extracellular matrix (ECM) and show similar behavior to epithelial cells.

The characteristics were considered for the tissue engineering for the biocompatibility and biodegradability presented by the PCL and the cyclodextrin. The nanofiber morphology was the same as ECM, and the diameter of the fibers was close to the diameter of the collagen and fibronectin fibers, which could be from 10 nm to some micrometers. In this regard, the theoretical study instructed us to correlate and explain the experimental results via computational modeling.

### 2.1. Molecular Parameters

The target materials were studied using computational chemistry methods. Previously to the optimization of the cyclodextrin complexes and their starting molecules, the conformer distributions of the built molecules were assembled, and the most stable conformers were selected (global minimum energy). Subsequently, the selected structures were appropriately optimized via DFT, employing the B3LYP functional and 6-31G(d,p) basis set. The structures obtained in phase gas are depicted in Figure 2 and Figure 3.

Regarding the optimized structures for the starter materials, the bond lengths for the PCL model were compared with the PCL single-crystal geometrical parameters reported by Chatani Y et al. [50]. The calculated values for the carbonyl group C=O were 1.237 Å in contrast to the experimental value of 1.20 Å. In this sense, the value of the C-C length bond as previously reported [50] was 1.54 Å, which was similar to the calculated value (1.539 Å). For β–cyclodextrin, the theoretical bond length of a primary alcohol (C6-O6) was compared with the data published by Linder and Saenger [51]. The calculated value was 1.442 Å compared with 1.430 Å from the experimental values. Finally, in the case of 6-deoxy-6-amino–β–cyclodextrin, the structure was like the structure of **β**–cyclodextrin, differentiated only by the amino group; in this sense, the calculated amine group carbon-nitrogen bond length was compared with the same bond of methyl benzyl ammonium salt [52], and the theoretical value was 1.456 in contrast with 1.484 Å. The rest of the theoretical values were coherent and near the experimental values, appropriated with the theory level used.

The structures of the inclusion complexes that formed the nanomaterials were also optimized at the same theory level. The optimized β–CD/PCL inclusion complex showed a channel-type structure [19,20]; however, the PCL was not localized in the center of the hydrophobic cavity (**the distance of PCL to the C6 carbons of glucose in cyclodextrin primary face was:** 5.060 Å, 4.613 and 4.240 Å). As a consequence of a hydrogen bond forming among a primary hydroxyl group of cyclodextrin and the carbonyl group of a PCL model, the bond length of O-H•••O=C was 1.795 Å and it had a bond angle of 169.1°; in this sense, this hydrogen bond was classified as being a strong bon [53]. A similar situation was observed in β–CDNH_2_/PCL, where the PCL was not centered in the hydrophobic cavity; however, the exhibited hydrogen bond was stronger than the present one in β–CD/PCL (-O-H•••O=C length at 1.753 Å and 176.48°). Finally, the EGF–β–CDNH_2_/PCL model showed a geometrical parameter in the same range as that of β–CDNH_2_/PCL; however, the hydrogen bond distance was bigger than β–CD/PCL and β–CDNH_2_/PCL, achieving a bond with a lower strength than the other 2 complexes (1.829 Å).

The electronic parameters of the optimized molecules (Table 1) were used to calculate the interaction energies among the cyclodextrin and the PCL; the results show that EGF– β–CDNH_2_/PCL exhibited lower interaction energy (−17.1 kcal/mol), followed by β–CDNH_2_/PCL (−20.9 kcal/mol) and finally β–CD/PCL (−60.6 kcal/mol), with higher interaction energy.

Another interesting property calculated was the dipole moment of the complexes and the starting materials. The PCL model showed less of a dipole moment (1.5227 Debye), taking into account the nonpolar bonds present and the hydrophobic character of the molecule; in the cases of β–cyclodextrin and amino β–cyclodextrins, the dipole moments were in the range of 8.20 and 10.33 Debye in the gas phase, being more polar than beta cyclodextrin. However, in the case of the complexes, β–CDNH_2_/PCL exhibited more polarity (5.9249 Debye) than the complex with β–CD (5.0998 Debye). The wettability, an important parameter for biomedical applications and measured using the contact angle, is dependent on the polarity of the material. In this sense, the contact angle values of the studied molecules were reported in a previous piece of work [47,48] for the fibers of PCL, β–CD/PCL, β–CDNH_2_/PCL and EGF–β–CDNH_2_/PCL and were 135.4° ± 2.18°, 124.5° ± 0.44°, 1.3° ± 0.61° and 1.4° ± 0.78°, respectively. These results showed the same tendencies as the dipole moment values of PCL: the less polar molecules exhibited less wettability; meanwhile, β–CDNH_2_/PCL displayed a bigger dipole moment, exhibiting more wettability [47,48]. The model of EGF–β–CDNH_2_/PCL showed a low dipole moment, since only the join chain and small moiety to the EGF protein (EGF model) was used, in comparison to the high number of polar groups in the protein. It is important to highlight that all of the complexes showed negative values of interaction energy and, consequently, spontaneous formation. Finally, the change in the dipole moment of the isolated molecules in comparison with the complexes was attributed to the polarity change in the cyclodextrin cavity after the PCL entered it [54].

### 2.2. Vibrational Study

The theoretical calculations for the vibrational modes using the infrared and the Raman spectrophotometric techniques were performed for the most stable conformer of the studied molecules and their starting materials. The obtained data were compared with the corresponding experimental values to make a detailed analysis of the corresponding vibrational modes.

The vibrational spectra calculations were achieved for free molecules in the gas phase. The vibrational absorption assignments were performed by comparing the molecules reported in the literature and those results obtained from the respective theoretical calculations [47,48,55,56]. The descriptions of the presented modes were approximated, and it is convenient to note that some of the vibrations were mixed. Hence, at the first stage, in the infrared spectrum analysis of the starting materials, the PCL model consisted of 59 atoms, with 171 normal vibrational modes (see Figure 4c). A total of 115 of these normal vibrations were distributed in-plane and 56 were out-of-plane, considering β–CD and β–CDNH_2_. As seen in Figure 4a,b, these molecules have 147 and 148 atoms, showing 435 and 438 normal vibrational modes, respectively, with 291 and 293 in-plane and 144 and 145 out-of-plane for β–CD and β–CDNH_2_, respectively.

Regarding the IR band results from the PCL model (Figure 4c), the bands presented in the range of 3146–2998 cm^−1^ were attributed to C-H stretching for the aliphatic chain contrasted with the experimental bands of 2930–2850 cm^−1^. Additionally, a band at 1685 cm^−1^ corresponds to a carbonyl group (C=O) of the PCL model ester group compared to the carbonyl experimental band of 1732 cm^−1^ [47,57,58].

β–CD and β–CDNH_2_ (Figure 4a,b) exhibited bands at high frequencies, corresponding to the OH group stretching of glucose molecules in the cyclodextrin at 3719–3349 cm^−1^ in both molecules. However, in the case of β–CDNH_2_, two additional bands are shown at 3690 and 1681 cm^−1^, attributed to N-H stretching and bending (scissoring), and are in agreement with the experimental results (around 3500 cm^−1^ and 1643 cm^−1^, respectively) [59,60,61,62]. The next bands were attributed to C-H stretching frequencies that appeared in the range of 3135–2984 cm^−1^, corresponding to the aliphatic C-H asymmetric stretching of the glucose hydrocarbonated chain. In the experimental spectra, the symmetrical and asymmetrical stretching bands were observed in the range of 2900–2800 cm^−1^, respectively. The scaled theoretical frequencies and the experimental data agreed [47,48].

The biomaterial models β–CD/PCL, β–CDNH_2_/PCL and EGF–β–CD-NH_2_/PCL (Figure 4d–f) had 206, 207 and 238 atoms with 612, 615 and 708 normal vibrational modes, respectively. Of these normal vibration modes, 409, 411 and 473 modes, respectively, were in-plane; meanwhile, 203, 204 and 235 vibrations were out-of-plane, correspondingly. Respecting the infrared bands of these molecules, the calculated infrared spectrum showed that sugar hydroxylic bands showed a shift to a high frequency (3780–3211 cm^−1^) [59,60,61,62] in comparison with the free molecules (3719–3349 cm^−1^) and the experimental results (broad bands around 3400 cm^−1^). The aliphatic C-H stretching was in the range of 3146 to 2967 cm^−1^, like with that observed in the experimental results. Finally, the presence of 2 bands corresponding to the carbonyl group of PCL, and the carbonyl group of PCL with a hydrogen bond with the cyclodextrin moiety, was presented in the range of 1780–1615 cm^−1^, agreeing with the presence of these bands in an experimental spectrum and showing the validity of the molecular models proposed in this work [47,48].

Theoretical Raman spectra of the studied materials were also calculated at the same theory level; see Figure 5.

The PCL model spectrum displayed the same bands observed in the experimental spectrum previously reported [47,48,58]. A broad band corresponding to C-H stretching (2900–3000 cm^−1^), in comparison with some bands in the theoretical spectra around 3157–3007 cm^−1^, showed an overestimation. Additionally, the carbonyl group showed a weak band: 1700 cm^−1^ in the experimental spectra and 1720 cm^−1^ in the theoretical calculation. Regarding β–CD and β–CDNH_2_, similar bands were observed in the range of 400–1500 cm^−1^ corresponding to a C-O-C glucose stretch, and 3519–3616 cm^−1^ as assigned to hydroxyl groups; these results agree with the experimental results [47,61]. In addition, β–CDNH_2_ displays two additional bands at 3696 cm^−1^ and 1735 cm^−1^ assigned to the N-H bond; in this case, both bands were overestimated (around 3100 and 1600 cm^−1^) [47,48].

The calculated Raman spectra of 3 novel complexes showed the bands characteristic of cyclodextrin C-O-C glucose stretching (400–1500 cm^−1^). However, EGF–β–CDNH_2_/PCL revealed an additional band related to L-valine bond stretching (955 cm^−1^). The carbonyl groups’ bands in the 3 novel materials were located around 1646–1720 cm^−1^, exhibiting 2 weaker bands that agreed with the experimental results (around 1700 cm^−1^) previously reported [47,48,63,64]. The vibrational results also allowed for the calculation of thermodynamic parameters, as seen in Table 2, from the inclusion process in the gas phase.

Thermochemical parameters revealed that the β–CDNH_2_/PCL complex demonstrated a lower formation of Gibbs free energy (−3.5294 kcal/mol), followed by EGF–β–CDNH_2_/PCL (−5.6117 kcal/mol) and finally β–CD/PCL (−47.3610 kcal/mol). The lower complex stability of β–CDNH_2_/PCL can be explained by considering the media in which the complexes were formed. The calculation was carried out in the gas phase, where the less polar molecules are more stable; as a consequence, the β–CD/PCL formation is more spontaneous. However, considering that the evaluated materials were previously studied using experimental methods in solid nanofiber form, the solvation effects were not considered.

### 2.3. Molecular Reactivity

The structure–activity relationship of the studied molecules was analyzed using the reactivity parameters calculated via quantum chemical methods, the reported cell viability of the target biomaterials (MTTs) [48] and the corresponding fibroblast cellular counts (values shown in Figure 6) [48]; additionally, the values of adsorption, metabolism and toxicological and physicochemical properties were predicted using cheminformatics tools.

GAP energy has been used to describe the reactivity and biological activity of several molecules [29,31,32,65,66,67,68]; in this case, the calculated gap values (Energy LUMO-Energy HOMO) of the studied molecules are displayed in Figure 7. These values show that the PCL model, with an ΔEgap = 169.0446 kcal/mol, is the least reactive (the most stable) molecule. The higher stability in the PCL model can be explained considering its big aliphatic chain and the fact that it is without reactive groups; additionally, this molecule had less cell viability and a lower cell count (MTT = 76% and cell count = 20,000). The most reactive compound was β–CDNH_2_/PCL, which was the molecule with the smallest energy gap (Egap = 120.7282 kcal/mol), but consequently the highest reactivity and higher cell viability (MTT = 83%, cell count = 35,000 cells). This material is more reactive than the PCL and β–CD/PCL and has more biological activity. These results suggest a possible correlation between the chemical stability of the material and cell proliferation. In relation to EGF–β–CDNH_2_/PCL, it showed less reactivity than β–CDNH_2_/PCL; however, its cell proliferation was higher in comparison with β–CDNH_2_/PCL. The difference in the behavior can be explained considering that EGF–β–CDNH_2_/PCL contains only a model of the EGF molecule; consequently, the reactivity can be different from the experimental molecule.

The value of Energy HOMO (E_HOMO_) is associated with the electron donating skill, and a higher value of E_HOMO_ is indicative of a greater easiness of donating electrons to unoccupied orbitals. A lower value of E_LUMO_ is related to the ability of the molecule to accept electrons. The quantum chemical parameters calculated via DFT are summarized in Table 3. In this sense, it is convenient to highlight that the E_HOMO_ energy values of PCL and β–CD/PCL (−165.2734 and −155.3905 kcal/mol, respectively) were greater than β–CDNH_2_/PCL and EGF–β–CDNH_2_/PCL (−133.0207 and −150.8978 kcal/mol), indicating that the presence of an amine group increases the electron donating capability and suggests a direct correlation between the electron donating capability and the cell viability, in agreement with the previously reported experimental results [48].

Additionally, the reactivity characterization of PCL, β–CD, β–CDNH_2_ and the three inclusion complexes continued via the calculation of the following parameters: DFT global chemical reactivity descriptors, hardness (η), chemical potential (μ) and electrophilicity (ω); these parameters were obtained considering the electronic affinity (EA) values and their first ionization energy (I), being calculated by the energy difference from the neutral, positive and negative structure for each compound of this study at the same theory level. The results are summarized in Table 4.

The energy values of the studied molecules, both in neutral and ionic forms, showed that the neutral form was the most stable and the positive form was less stable. The electronic affinity values were, in all cases, positive, indicating a lower trend to accept electrons, but the molecule that released the least energy to form an anionic structure was β–CDNH_2_/PCL (0.0381 Hartree) and was the more reactive one. In the same way, β–CD has higher energy (0.0708 Hartree), being less reactive. Regarding the first ionization energy, β–CDNH_2_/PCL showed lower energy (0.272 Hartree) and higher reactivity, followed by EGF–β–CDNH_2_/PCL (0.276 Hartree) and finally β–CD/PCL (0.285 Hartree), being less reactive than the PCL (0.325 Hartree); this behavior in terms of reactivity agreed with the displayed biological activity.

The negative chemical potential (μ) can be called the absolute electronegativity. Electronegativity shows how the electrons will flow from high-electronic-density regions in a molecule to another site of lower electronic density [69,70,71]. As such, this is an important index of reactivity. The absolute electronegativities, as seen in Table 4, are in the following order: PCL model > β–CD > β–CDNH_2_. This means that molecule PCL is more prone to attract electrons during the interaction with another chemical compound; in addition, the electronegativity of the inclusion complexes is in the following order: β–CD/PCL > EGF–β–CDNH_2_/PCL > β–CDNH_2_/PCL, showing more of an inverse order than for biological activity.

Electrophilicity and Lewis’s acidity are related [72]. It has been shown that electrophilicity possesses information regarding structure, stability, reactivity, toxicity, bonding interactions, etc. The global electrophilicity index measures the stabilization energy when an optimal electronic charge transfer from the environment to the system occurs [73]. A small ω-value indicates that the compound can be designated as being nucleophilic. Regarding the study molecules, β–CDNH_2_/PCL and EGF–β–CDNH_2_/PCL showed lower values of electrophilicity and consequently nucleophilic behavior.

Considering previous reports, it has been described that the direct correlation between the electrophilic character of diverse molecules with toxicity produces the most facile binding partner to the drug receptor and other biological activities [74,75,76]. These results highlighting the electrophilicity, first ionization energy and chemical potential are useful descriptors of the biological activity in this kind of material.

The contours of the frontier molecular orbitals for the studied molecules are displayed in Figure 8 and Figure 9. In the cases of β–CD/PCL and β–CDNH_2_/PCL, the LUMO orbitals were located on the PCL carboxyl group and were associated with the electrophilic character of carbon in the carboxyl group. Meanwhile, EGF–β–CDNH_2_/PCL showed a LUMO surface on amino acid residue. Regarding HOMO orbitals, β–CD/PCL was located on one glucose unit, which did not form a hydrogen bond with PCL. Similar behavior was observed in the β–CDNH_2_/PCL material, where the HOMO orbital was in the glucose unit with the amine group; in both cases, the hydroxyl and amine groups were nucleophilic. Finally, the HOMO orbital of EGF–β–CDNH_2_/PCL was situated in the hydrocarbon chain which connects β–CDNH_2_/PCL with the amino acid L-valine (EGF model).

### 2.4. Molecular Electrostatic Potential Maps

Using an electrostatic potential map is an appropriated method to suitably predict molecular reactivity in addition to achieving biological studies of interesting compounds [77]. Furthermore, this property has been considered as an indicator of the reactivity regions of a target molecule; therefore, it has been employed to study electron–donor and electron–acceptor interactions, for example, among bioactive compounds and cellular receptors [78]. Thus, this property was calculated for the three studied molecules at a level of theory of B3LYP/6-31G(d,p) and is displayed in the corresponding maps by a color range from 52.00 kcal (deepest blue) to −46.00 kcal (deepest red), as shown in Figure 10.

Figure 10 presents the nucleophilic sites in red and the blue color shows the electrophilic sites; moreover, the potential values are shown in the highest and lowest electron density sites. A higher electronic density was seen in the oxygen atoms of the hydroxyl groups (value range of −45.55 to −32.33 kcal/mol) and in a minor distribution in the carboxyl group oxygens (potential values between −42.95 and −38.48 kcal/mol). In the case of β–CDNH_2_/PCL, the amine group also showed a high value (−14.98 kcal/mol). In addition, the higher electronic deficiencies in the material are located in the hydrogen atoms of the hydroxyl groups which form hydrogen bonds (31.89 to 14.14 kcal/mol). Additionally, EGF–β–CDNH_2_/PCL exhibits another electron density deficiency in the hydrogen atom of the amino acid residue used as the EGF model (52.15 kcal/mol).

Concerning the cheminformatics results, the first analysis was made by the PASS server, which can be used for the prediction of different types of pharmacological activities of different substances, and is based on a modified naïve Bayes algorithm; it was applied since it has been shown to be robust and to provide good predictions of many biological activities based on just the structural formula of a compound [80,81]. In this sense, the models for PCL, β–cyclodextrin (β–CD) and 6-deoxy-6-amino beta-cyclodextrin (β–CD-NH_2_) were individually analyzed in the server. It is convenient to note that EGF–β–CD-NH_2_ was not analyzed by this server because the grafted protein structure cannot be processed by the algorithm to the server. The obtained results are summarized in Table 5.

The results show that β–CD and β–CDNH_2_ exhibit a high probability of anti-inflammatory activity (0.962 and 0.918, respectively). These results showed promising skin wound-healing activity for this material, which has been observed in experimental results. Other works report a good experimental anti-inflammatory effect with lesser Pa values obtained in comparison with the prepared material [35]. It is also convenient to highlight that some other properties of the wound-healing material are the addition of anti-inflammatory and analgesic compounds, as well as growth factors [82,83]; in this sense, the presence of the cyclodextrins in the studied materials improves the wound-healing properties in contrast with only PCL.

Regarding the complementarity with the PASS server results, some toxicological and physicochemical properties of the starting materials were used to evaluate some bioactive properties. The prediction of their toxicity risk and some significant physicochemical properties was performed using the OSIRIS-Property-Explorer algorithm and Osiris-Data warrior [31,32,35,84,85]; the corresponding results are summarized in Table 6.

The toxicity risk predictor shows that the studied structures, β–CD and β–CD-NH_2_, had less of a risk of undesirable effects; they do not present risks of mutagenicity, tumorigenicity, irritant or reproductive effects in comparison with the PCL model, which presents a high risk of an irritant effect. These results indicated that the risk is due to the presence of the long aliphatic chain, explaining why β–CD and β–CD-NH_2_ do not present this effect.

Several physicochemical properties of the compounds were also estimated, as seen in Table 6, such as cLog *P*, whereby the logarithm of its partition coefficient between *n*-octanol and water is a property that describes the molecular hydrophobicity. The values of β–CD and β–CD-NH_2_ for this property were −12.86 and −13.257, respectively, in the studied compounds (<5) [32,35,55,86,87]. In this sense, the obtained values were less than 5, indicating a reasonable probability of being well absorbed [85,88,89]. Meanwhile, the PCL model presented lower values (>5), showing the high hydrophobicity. These results, in addition to those observed in quantum chemistry calculations, confirm the increase in hydrophilicity of the material when the complexes are formed.

Drug solubility (expressed as cLog *S*) is an important factor in describing the absorption process. Poor solubility leads to poor absorption and bioavailability [31,86,87,89]. The commonly traded drugs have cLog *S* values of greater than −4. It is important to highlight that the best solubility is observed for β–CD and β–CD-NH_2_, also indicating that these compounds possess the best absorption, movement in the bloodstream and better disposal via the urinary tract after the materials are metabolized. In this sense, the PCL model had lower solubility, showing a low absorption value (cLog *P* = 6.4546 and cLog *S* = −4.57); this result is in agreement with less wettability, as reported in other works [85,88].

The drug score (DS) is the combination of drug-likeness, cLog *P*, cLog *S*, molecular weight and toxicity risks in one handy value that can be used to judge the compound potential to qualify as a drug [90]. In this sense, β–CD and β–CD-NH_2_ presented similar drug scores, both being higher than the PCL model. The lower drug score predicted for PCL was a consequence of the high probability of an irritant effect in these molecules [31,32].

Complementary to the results obtained are the predictions of the absorption, metabolic and excretion properties of the substrates or materials. For the studied molecules in human metabolism, the pseudo rotaxane dissociation in metabolic processes was considered. The different behaviors in both the absorption and the excretion models were predicted using admetSAR methodology, and the corresponding results are exhibited in Table 7. The predictions for PCL, β–CD and β–CDNH_2_ showed that only PCL exhibits human intestinal absorption, with a probability value of 0.949, whereas β–CD and β–CDNH_2_ do not experience absorption. Concerning the blood–brain barrier, only the PCL model showed a good absorption probability (0.954).

Regarding Caco-2 (human colon adenocarcinoma) permeability, this parameter is considered to be the “gold standard” for drug permeability accepted by the FDA [81,91]. In this sense, the prediction for the target molecules exhibited permeability only for PCL. The target molecules transported by P-glycoprotein, which is one of the principal transporters involved in the clearance of xenotoxins [92,93,94], showed no possibility of being a substrate or inhibitor of this protein and, consequently, the bioavailability of all of the components of the studied materials could be minor. It is convenient to highlight that the materials were designed to act at the level of the skin and the transport of the molecules by this protein should not affect its biological activities.

## 3. Materials and Methods

### 3.1. Optimization

For the epithelial growth factor (EGF) anchored to β–CDNH_2_/PCL, it was required that we establish a model considering the complex of these structures. The molecular protein model (EGF) used was the L-valine amino acid, considering that in the experimental obtained material, the protein was bound to β–CDNH_2_/PCL via the imine groups formed with glutaraldehyde, as indicated in Figure 1.

In the first stage, the PCL and EGF models were built considering standard bond lengths and bond angles, using the PC Spartan 06 program [95] followed by conformer distribution analysis and the selection of the most stable conformers of each molecule; for β–cyclodextrin and its derivative, the three-dimensional crystal structures were obtained from Cambridge Crystallographic Data Centre (CCDC) [96,97].

The selected geometries were appropriately optimized using the Gaussian 09 program [98]. Density functional theory (DFT) calculations were also employed [99]. Becke’s three-parameter hybrid density functional, B3LYP [100,101] and 6–31G(d,p) basis set including the split-valance and polarized functions [102,103] was used. Since the experimental novel materials’ characterization previously reported was made according to solid nanofibers without the use of solvents, the calculations were carried out without the solvation effect.

### 3.2. Interaction Energy Calculations

The interaction energies (E_interaction_) of the studied materials were calculated by subtracting from the corresponding inclusion complex total energy (E_complex_) the total energy of its isolated and completely relaxed constituting molecules (E_CD_ and E_PCL_), as indicated in Equation (1) [33,47,104].
(1)$$E_{\mathrm{interaction}}=E_{\mathrm{Complex}}-E_{\mathrm{CD}}-E_{\mathrm{PCL}}$$

### 3.3. Vibrational Analysis

Vibrational frequencies calculated using density functional theory offered excellent results in organic compounds, when the calculated frequencies were scaled to compensate for the approximate treatment of electronic correlation, for basis set deficiencies and for anharmonicity [105,106,107,108]. In this sense, Rauhut and Pulay [109] obtained the vibrational spectra of a few molecules using B3LYP with the 6–31G(d) basis set, which were used to decrease the uncertainties in assignments in the infrared spectra. Therefore, in this work, using the DFT/B3LYP/6–31G(d,p) method, the corresponding vibrational frequencies were calculated. Regarding the quantum chemical literature [55], the B3LYP functional yields a good description of harmonic vibrational wavenumbers. However, the quantum chemical results differ from the measured ones and, consequently, these calculations were scaled; in this sense, the thermal contributions to the vibrational energy were scaled by 0.963 [43,44].

### 3.4. Thermochemical Parameters

Thermochemical values were estimated from frequency calculations, including a thermochemical analysis of the system considering 298.15 K, 1 atm of pressure and the principal isotope for each element, all at the same theory level. Zero-point correction to the electronic energy (ZPE) of the molecule was used to calculate the values of enthalpy and Gibbs free energy (Equations (2) and (3)) [31,33,44,110].
(2)$$∆H_{\mathrm{process}}=H_{\mathrm{products}}-H_{\mathrm{reactants}}$$
(3)$$∆G_{\mathrm{process}}=G_{\mathrm{products}}-G_{\mathrm{reactants}}\;$$
where the ΔH_process_ corresponds to the enthalpy change in the inclusion process, and H_products_ and H_reactants_ were the formation enthalpy of products (complex) and reactants (Cyclodextrins and PCL); in this sense, the ΔG_process_ was the Gibbs free energy involved in the inclusion process. Meanwhile, the G_products_ and G_reactants_ were the Gibbs free energy of inclusion complexes (products) and the reactants (cyclodextrins and PCL), respectively.

### 3.5. Reactivity Parameters

The highest occupied molecular orbital–lowest unoccupied molecular orbital (HOMO−LUMO) gap is a typical quantity used to describe the dynamic stability of molecules (Equation (4)) [31,111]. Values of the orbital energy and the surface of the frontier orbitals were calculated using the same level of theory (B3LYP/6-31G(d,p)). In addition, the molecular electrostatic potential maps (MEPs) were obtained for the target molecules to complete the electronic analysis, considering the importance of these results in the interaction models in the studied biological system [78].
(4)$$E_{\mathrm{gap}}=E_{\mathrm{LUMO}}-E_{\mathrm{HOMO}}$$
where E_gap_ corresponds to the gap energy, and E_LUMO_ and E_HOMO_ are LUMO and HOMO molecular orbital energy.

### 3.6. Reactivity Descriptors

The chemical potential (μ), hardness (η) and electrophilicity (ω) were calculated using their respective equation [72,112,113,114]:(5)$$\mu =-\frac{\left(I+\mathrm{EA}\right)}{2}$$
(6)$$n=\frac{\mathrm{IP}-\mathrm{EA}}{2}$$
(7)$$\omega =\frac{\mu^{2}}{2n}$$
where I corresponds to the ionization potential and EA corresponds to electron affinity; these values were calculated from the energies of the neutral, cationic and anionic forms of the complexes and the isolated molecules at the same theory level.

### 3.7. Biological Activity Predictions

The biological activity predictions were made using the chemoinformatic PASS server, which predicts over 3500 kinds of biological activities. The predicted biological activity profile was obtained from the structural formula of compounds and was based on the analysis of structure–activity relationships for more than 250,000 biologically active substances, including drugs, drug candidates, leads and toxic compounds [35,80,115,116,117].

### 3.8. Toxicological, Physicochemical and Metabolic Properties Prediction

The physicochemical properties predicted for the molecules under study provide pertinent information about the ability of a molecule to interact with the amino acid residues inside the cells or the membrane receptors. The toxicological risk and physicochemical properties of the studied molecules were obtained using the Data warrior OSIRIS property explorer. The toxicological risk prediction process relies on a precompiled set of structural fragments that give rise to toxicity alerts in case they are encountered in the structure currently drawn. cLog *P* and cLog *S* were estimated using the OSIRIS method, which was implemented as a system for adding the contributions of every atom based on its properties. The drug-likeness approach is based on a list of about 5300 distinct substructure fragments with associated drug-likeness scores. The drug-likeness was calculated by employing the score values of the fragments present in the molecule under investigation [84,88]. The absorption and metabolic properties of the studied compounds were calculated using the admetSAR server [81,92,118], which predicts about 50 ADMET endpoints using a chemoinformatics-based toolbox called ADMET-Simulator, which integrates high-quality and predictive QSAR models.

## 4. Conclusions

An in silico study, based on quantum chemistry and chemoinformatics methods for three novel materials, was carried out; it is important to note that the obtained theoretical results agree with the data reported in experimental studies previously performed by our research group.

First, the physicochemical properties such as the wettability can be correlated with the dipole moment. The results showed that β–CD-NH_2_/PCL and EGF–β–CD-NH_2_/PCL exhibited higher polarity and consequently a lower contact angle. Additionally, the best stability of the complex β–CD-NH_2_/PCL over β–CD/PCL was corroborated by the interaction energy and thermochemical parameters. In addition, theoretical spectroscopic data agreed with the experimental results as part of this chemical characterization.

The more bioactive materials were EGF–β–CD-NH_2_/PCL and β–CD-NH_2_/PCL, exhibiting the lowest energy gap of the HOMO and LUMO orbitals and consequently high reactivity values; meanwhile, the lowest cell viability was for PCL, showing the highest energy gap value. These results are indicative of a possible correlation between the chemical stability and bioactivity for this kind of material. The chemoinformatics results evidenced the anti-inflammatory effect of β–cyclodextrin and its derivatives, favoring the explanation of the corresponding in vivo results, as previously reported.

This work contributes to the theoretical characterization of biomaterial derivates of β–cyclodextrin inclusion complexes; thus, taking into account the obtained correlations, many other related materials can be designed to improve the knowledge about their bioactivity. A future perspective of this work is to determine experimentally and theoretically the biodegradability, the effects of the position of PCL in the β–CD, β–CDNH_2_ and EGF–β–CDNH_2_ cavities surrounding the complex dipole moment of complexes and, finally, the effects of solvation regarding this class of materials.

## Figures and Tables

**Figure 1 ijms-24-08932-f001:**
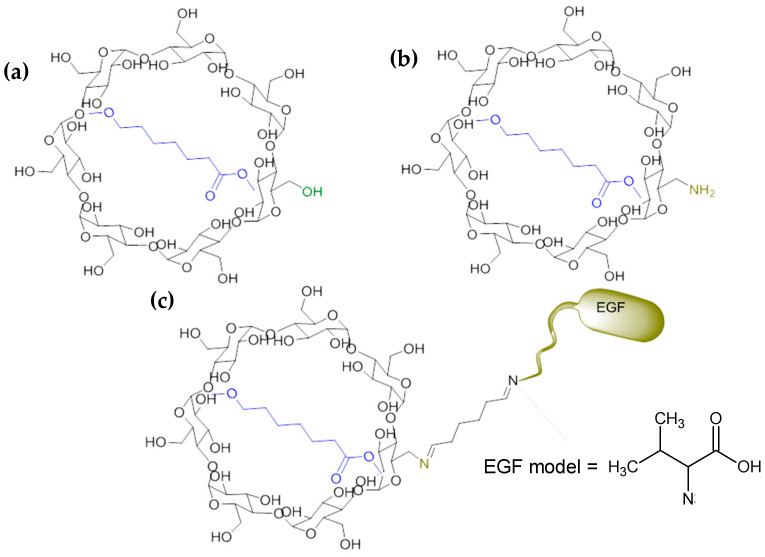
Schematic representation of studied inclusion complexes (**a**) **β**–CD/PCL, (**b**) **β**–CDNH_2_/PCL and (**c**) EGF–**β**–CDNH_2_/PCL and the EGF model used in this work.

**Figure 2 ijms-24-08932-f002:**
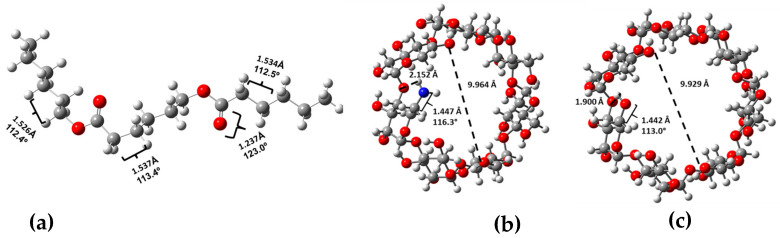
Starter material optimized structures with selected geometrical parameters (bond lengths are in Å, bond angles are in degrees and the dashed line corresponds to cavity diameters in Å) for (**a**) PCL, (**b**) β–CDNH_2_ and (**c**) β–CD.

**Figure 3 ijms-24-08932-f003:**
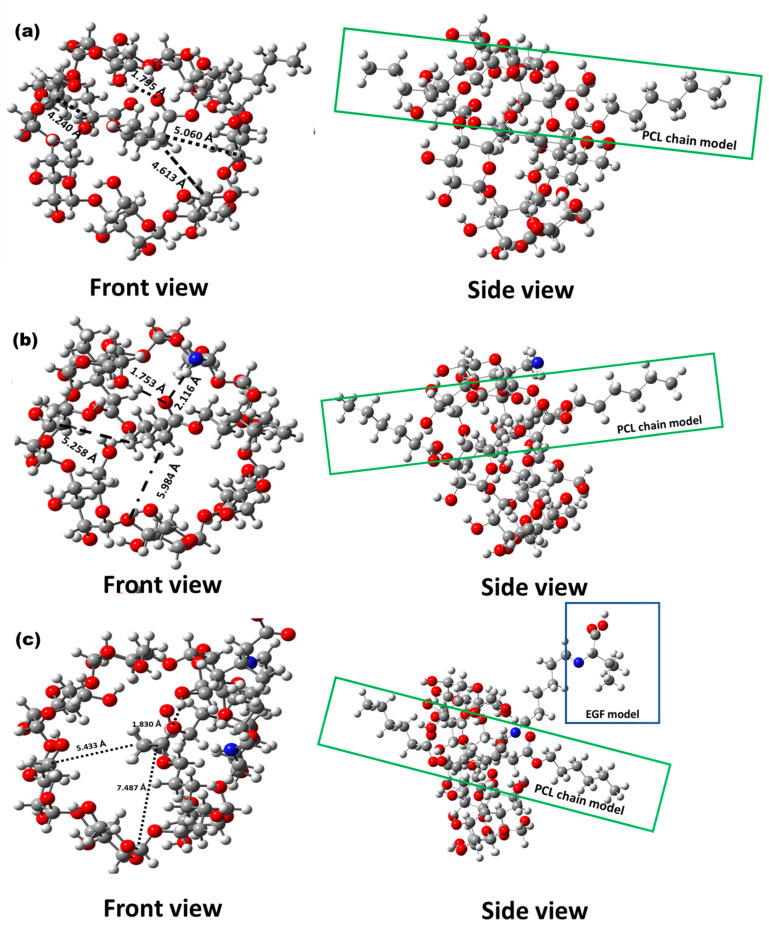
Optimized structure of the complexes from front and side views, with selected geometrical parameters (bond lengths are in Å, bond angles are in degrees and the dashed line corresponds to cavity diameters in Å): (**a**) β–CD/PCL, (**b**) β–CDNH_2_/PCL and (**c**) EGF–β–CDNH_2_/PCL.

**Figure 4 ijms-24-08932-f004:**
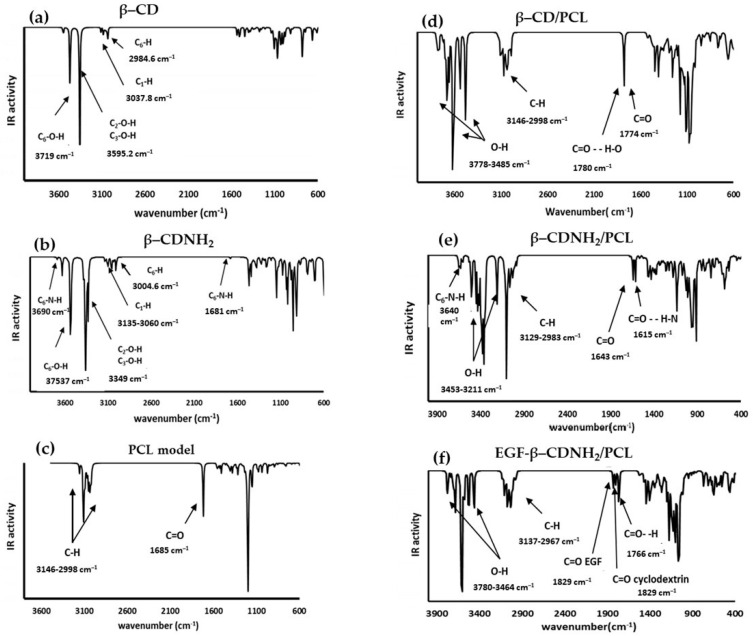
Calculated infrared spectra of (**a**) β–CD, (**b**) β–CDNH_2_, (**c**) PCL, (**d**) β–CD/PCL, (**e**) β–CDNH_2_/PCL and (**f**) EGF–β–CDNH_2_/PCL. Experimental spectra were reported in [47,48].

**Figure 5 ijms-24-08932-f005:**
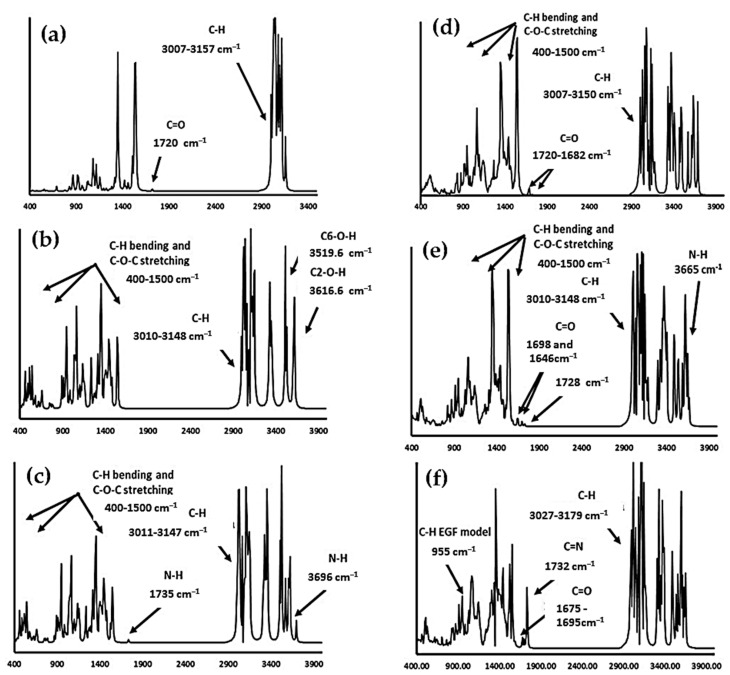
Calculated Raman spectra of (**a**) PCL, (**b**) β–CD, (**c**) β–CDNH_2_, (**d**) β–CD/PCL, (**e**) β–CDNH_2_/PCL and (**f**) EGF–β–CDNH_2_/PCL. Experimental spectra were reported in [47,48].

**Figure 6 ijms-24-08932-f006:**
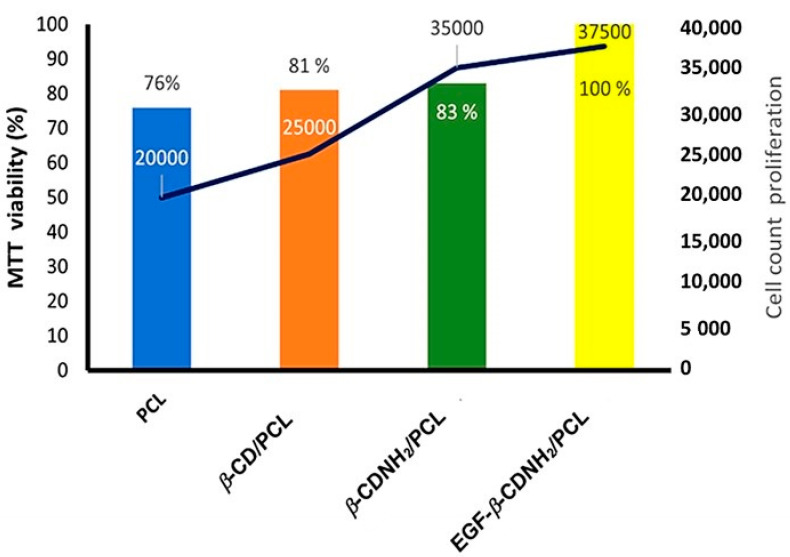
Fibroblast viability effect of target materials PCL, β–CD/PCL, β–CDNH_2_/PCL and EGF–β–CDNH_2_/PCL.

**Figure 7 ijms-24-08932-f007:**
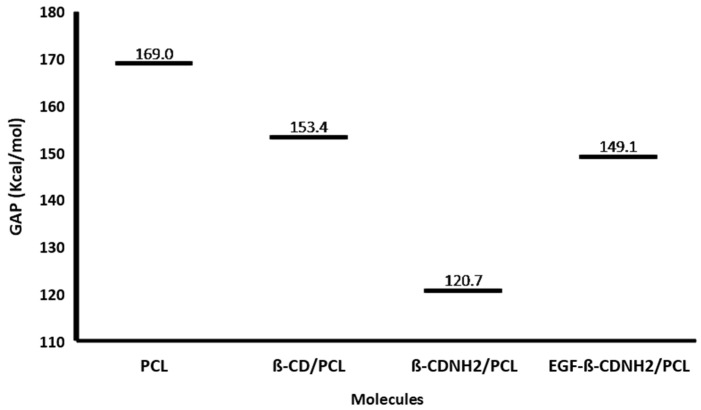
GAP values of target materials. PCL, β–CD/PCL, β–CDNH_2_/PCL and EGF–β–CDNH_2_/PCL.

**Figure 8 ijms-24-08932-f008:**
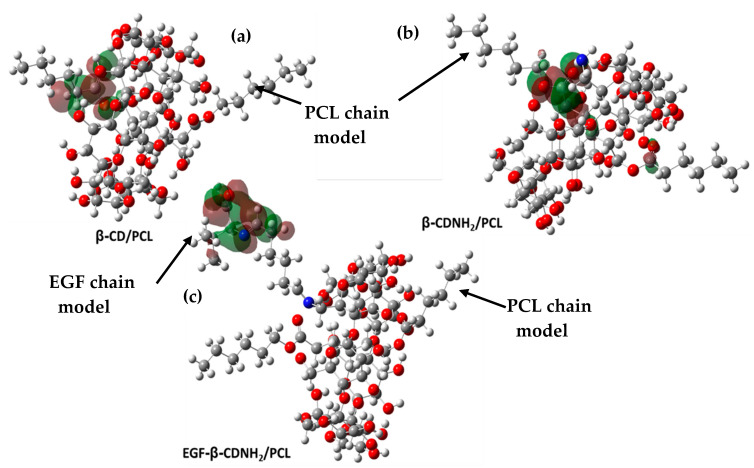
LUMO orbital surface of materials. (**a**) β–CD/PCL, (**b**) β–CDNH_2_/PCL and (**c**) EGF–β–CDNH_2_/PCL.

**Figure 9 ijms-24-08932-f009:**
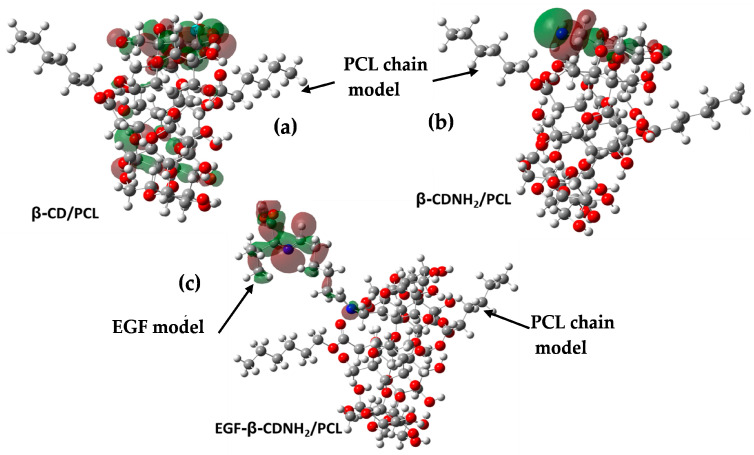
HOMO orbital surface of materials. (**a**) β–CD/PCL, (**b**) β–CDNH_2_/PCL and (**c**) EGF–β–CDNH_2_/PCL.

**Figure 10 ijms-24-08932-f010:**
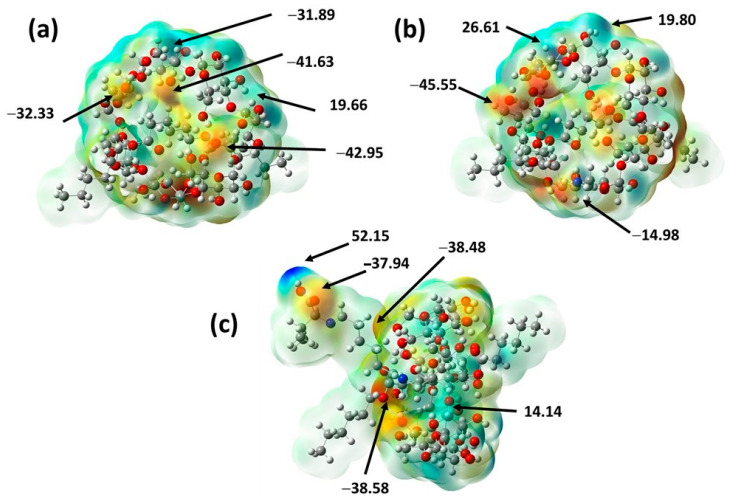
Molecular electrostatic potential surface of the studied materials, with selected values in kcal/mol [79] (**a**) β–CD/PCL, (**b**) β–CDNH_2_/PCL and (**c**) EGF–β–CDNH_2_/PCL.

**Table 1 ijms-24-08932-t001:** Electronic and interaction energy, and dipole moments of β–CD/PCL, β–CDNH_2_/PCL and EGF–β–CDNH_2_/PCL.

Electronic Parameters	Electronic Energy (Hartree)	Interaction Energy (E_interacction_) (Hartree)	Interaction Energy (E_interacction_) (kcal/mol)	Dipole Moments (Debye)
PCL	−1007.39355	N/A	N/A	1.5227
**β**–CD	−4275.32655	N/A	N/A	10.3339
**β**–CDNH_2_	−4255.47834	N/A	N/A	8.2063
**β**–CD/PCL	−5282.81672	−0.09662188	−60.6	5.0998
**β**–CDNH_2_/PCL	−5262.90531	−0.03341895	−20.9	5.9249
EGF–**β**–CDNH_2_/PCL	−5897.57936	−0.0272599	−17.1	3.2688

**Table 2 ijms-24-08932-t002:** Thermochemical properties of starting materials and inclusion complexes (kcal/mol).

Molecules	ΔH_ormation_ (kcal/mol)	ΔG_formation_ (kcal/mol)	ΔH_process_ (kcal/mol)	ΔG_process_ (kcal/mol)	Equilibrium Constant (Keq)
PCL model	−6.3139 × 10^5^	−6.3145 × 10^5^	N/A	N/A	N/A
β–CD	−2.6820 × 10^6^	−2.6821 × 10^6^	N/A	N/A	N/A
β–CDNH_2_	−2.6678 × 10^6^	−2.6679 × 10^6^	N/A	N/A	N/A
β–CD/PCL	−3.3117 × 10^6^	−3.3118 × 10^6^	−50.6876	−47.3919	5.5082 × 10^34^
β–CDNH_2_/PCL	−3.2992 × 10^6^	−3.2992 × 10^6^	−19.4832	−3.5309	3.8812 × 10^2^
EGF–β–CDNH_2_/PCL	−3.7203 × 10^6^	−3.7205 × 10^6^	−14.1420	−5.6117	1.3074 × 10^4^

**Table 3 ijms-24-08932-t003:** E_HOMO_ and E_LUMO_ of the studied molecules: PCL, β–CD/PCL, β–CDNH_2_/PCL and EGF–β–CDNH_2_/PCL.

	Energy (kcal/mol)		
Material	E_HOMO_	E_LUMO_	GAP (kcal/mol)
PCL	−165.2734	3.7712	169.0446
β–CD/PCL	−155.3905	−2.0017	153.3889
β–CDNH_2_/PCL	−133.0207	−12.2924	120.7282
EGF–β–CDNH_2_/PCL	−150.8978	−1.7758	149.1220

**Table 4 ijms-24-08932-t004:** Reactivity parameters of PCL, β–CD, β–CDNH_2_, β–CD/PCL, β–CDNH_2_/PCL and EGF–β–CDNH_2_/PCL.

Parameters		PCL	β–CD	β–CDNH_2_	β–CD/PCL	β–CDNH_2_/PCL	EGF–β–CDNH_2_/PCL
Energy (Hartrees)	Neutral	−1007.39355	−4275.32655	−4255.47834	−5282.81672	−5262.90531	−5897.57936
	Positive	−1007.06894	−4275.04279	−4255.20427	−5282.53199	−5262.63330	−5897.302952
	Negative	−1007.33909	−4275.2557	−4255.40896	−5282.77010	−5262.87433	−5897.541407
Reactivity parameters	EA	0.0545	0.0708	0.06934	0.047	0.031	0.0380
	I	0.325	0.284	0.274	0.285	0.272	0.276
	η	0.135	0.106	0.102	0.119	0.121	0.119
	μ	−0.190	−0.177	−0.172	−0.166	−0.151	−0.157
	Ω	0.133	0.148	0.144	0.115	0.095	0.104

**Table 5 ijms-24-08932-t005:** PASS server results: (**a**) PCL, (**b**) β–CD and (**c**) β–CDNH_2_.

(a)		(b)		(c)	
Pa	Biological Activity	Pa	Biological Activity	Pa	Biological Activity
0.953	All-*trans*-retinyl-palmitate hydrolase inhibitor	0.962	Anti-inflammatory	0.972	Pullulanase inhibitor
0.940	Sugar-phosphatase inhibitor	0.934	Sugar-phosphatase inhibitor	0.968	Glucan 1,4-alpha-maltotriohydrolase inhibitor
0.939	Alkenylglycerophosphocholine hydrolase inhibitor	0.905	UDP-N-acetylglucosamine 4-epimerase inhibitor	0.918	Anti-inflammatory
0.935	Alkylacetylglycerophosphatase inhibitor	0.901	Alkenylglycerophosphocholine hydrolase inhibitor	0.91	4-Alpha-glucanotransferase inhibitor
0.934	Acylcarnitine hydrolase inhibitor	0.898	Apoptosis agonist	0.906	Beta-amylase inhibitor
0.931	Cutinase inhibitor	0.866	Exoribonuclease II inhibitor	0.877	Amylo-alpha-1,6-glucosidase inhibitor

Pa = probability to activity.

**Table 6 ijms-24-08932-t006:** OSIRIS toxicological and physicochemical predicted properties of PCL, β–CD and β–CD-NH_2_.

		β–CD	β–CD-NH_2_	PCL Model
Physicochemical properties	cLog *P*	−12.86	−13.257	6.4546
	cLog *S*	1.57	1.494	−4.57
	Drug-likeness	−10.037	−9.5768	−27.332
	Drug score	0.2499585	0.24995251	0.13383782
Toxicity risks	Mutagenic	N	N	N
	Tumorigenic	N	N	N
	Reproductive effects	N	N	N
	Irritant	N	N	H

N = no risk, M = medium risk, H = high risk.

**Table 7 ijms-24-08932-t007:** Prediction of the absorption of the molecules in different models: (a) PCL, (b) β–CD and (c) β–CDNH_2_.

Absorption Model	(a)		(b)		(c)	
	Results	P	Results	P	Results	P
Blood–brain barrier	BBB^+^	0.954	BBB^+^	0.584	BBB^−^	0.903
Human intestinal absorption	HIA^+^	0.949	HIA^−^	0.814	HIA^−^	0.863
Caco-2 permeability	Caco2^+^	0.659	Caco2^−^	0.775	Caco2^−^	0.733
P-glycoprotein substrate	Non-substrate	0.604	Non-substrate	0.567	Non-substrate	0.571
P-glycoprotein inhibitor	Non-inhibitor	0.849	Non-inhibitor	0.886	Non-inhibitor	0.873
	Non-inhibitor	0.879	Non-inhibitor	0.981	Non-inhibitor	0.961
Renal organic cation transporter	Non-inhibitor	0.879	Non-inhibitor	0.818	Non-inhibitor	0.820

P = probability.

## Data Availability

All data was include in this manuscript.

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
