# Peer review of "In Silico Study of Novel Cyclodextrin Inclusion Complexes of Polycaprolactone and Its Correlation with Skin Regeneration"

_ijms, 2023, doi:10.3390/ijms24108932_

Round 1
Reviewer 1 Report
The manuscript reported three biomaterials obtained and its starter materials were characterized by theoretical methods in detail. It can be published after minor revision.
1. The language can be improved to make it easier to understand.
2. Why is the baseline of IR so straight?
Author Response
Reviewer: The language can be improved to make it easier to understand.
Us: Done, the paper was carefully revised according to the suggestion, please see the corrected manuscript
Reviewer: Why is the baseline of IR so straight?
Us: Since the infrared spectra were obtained by theoretical methods, hence the baseline settles straight.
Reviewer 2 Report
In this manuscript, the results of this research are conveyed thoughtfully and completely, and they are consistent with the experimental findings. However, the authors failed to explain and draw out the novelty of the work, this aspect needs to be improved. This work is worthwhile to be publish in this journal after minor revision. The following issues should be addressed:
1. Introduction is well-organized but the importance and novelty of the research should be highlighted and more clearly stated. The authors should give some examples of works in the bibliography, to clear the advantage of their work in comparison with those works.
2. Maybe the author should compare their results clearly with other reported works, highlighting the advantage and disadvantages of their novel composite.
3. The authors are responsible for the English, which should be polished throughout the manuscript to clear some minor typo/grammar errors.
4. Introduction part, if possible, some important and relative reports references could help:
https://doi.org/10.3390/ma16062170
https://doi.org/10.1007/s10904-023-02604-0
https://doi.org/10.1016/j.est.2023.107168
https://doi.org/10.3390/chemengineering7010004
https://doi.org/10.3390/ma15134547
https://doi.org/10.1016/j.ceramint.2022.05.151
Author Response
Reviewer: Introduction is well-organized, but the importance and novelty of the research should be highlighted and more clearly stated. The authors should give some examples of works in the bibliography, to clear the advantage of their work in comparison with those works. Introduction part, if possible, some important and relative reports references could help:
https://doi.org/10.3390/ma16062170
https://doi.org/10.1007/s10904-023-02604-0
https://doi.org/10.1016/j.est.2023.107168
https://doi.org/10.3390/chemengineering7010004
https://doi.org/10.3390/ma15134547
https://doi.org/10.1016/j.ceramint.2022.05.151
Us: Done, we have improved the introduction, adding convenient commentaries related to the suggesting references; we acknowledge the referee notes.
Reviewer: The authors are responsible for the English, which should be polished throughout the manuscript to clear some minor typo/grammar errors.
Us: Done, the paper was carefully revised according to the suggestion, please see the corrected manuscript.
Reviewer 3 Report
The manuscript entitled " In silico study of novel cyclodextrins inclusion complexes of polycaprolactone and its correlation with skin regeneration", Discuss a significant point with a clear findings.
Some point need to be covered that will increase the valuablity of results
1 -Please support the ntroduction with some refernces that support the main idea and the targeted results, this will make the design of the research more clear.
2- Some points in the methodolgy need to write in more details.
3 - Please adjust the Figures as correct resolutions and in the right capture (for example Fig.5 s not adjust, and seemed cutt off).
4 - Please explain the finding of results and what may it link or explain.
5- The discussion was written as well as need.
6- Please make a recommendation for the conclusion part and if possible the future applications.
7- A fine check needed for language at few parts of the manuscript.
My recommendation is Minor revision before acceptance.
Author Response
Reviewer: Please support the Introduction with some references that support the main idea and the targeted results, this will make the design of the research more clear.
Us: Done, we improved conveniently the introduction, adding appropriated comments-references as suggested by the second referee.
Reviewer: Some points in the methodology need to write in more details.
Us: Done, we enhanced the methodology section, as it can be seen in the new manuscript.
Reviewer: Please adjust the Figures as correct resolutions and in the right capture (for example Fig.5 not adjust, and seemed cutt off).
Us: Done, the Figures have been suitably corrected providing better resolution, please see the new manuscript.
Reviewer: Please explain the finding of results and what may it link or explain
. Us: Done, please see the new manuscript.
Reviewer: The discussion was written as well as need.
Us: Done, the discussion was revised, accordingly to the suggestion
Reviewer: Please make a recommendation for the conclusion part and if possible, the future applications.
Us: Done, we added expected applications and future works
Reviewer: A fine check needed for language at few parts of the manuscript.
Us: Done, the paper was carefully revised according to the suggestion, please see the corrected manuscript
Reviewer 4 Report
In the manuscript authors investigate some physico-chemical properties of three inclusion complexes of beta-cyclodextrin (and its derivatives) with polycaprolactone. For this purpose authors use low-level DFT method B3LYP/6-31G(d,p) and some chemoinformatic tools. Authors also state that their results match well the experimental data.
First and the major problem is lack of any model of solvent in performed calculations. I assume that experiments were not performed in vacuum (if it is not true authors should clearly state in the manuscript that they compare to the gas phase experimental data) and therefore their usefulness is severely limited. Also comparison to the experimental data obtained in solvent is not justified. Especially free energies of complex formation are usually strongly affected by the presence of solvent.
Second major problem is related to the procedure of optimization and global energy minimum search (see point 3 below).
There are many problems with the manuscript:
1. (line 23) What's the point of reporting absolute electronic energies of complexes in the abstract? What kind of information do these quantities carry? IMO these values (strongly method-dependent) calculated for three different compounds carry absolutely no valuable information.
2. Authors state that one of the investigated complexes is EGF-betaCDNH2/PCL (abstract and line 55). What exactly is EGF. Some minimal information about it should be provided. I assume that it is small protein. If this is the case - do authors really performed DFT calculations of complex with this small protein? If yes what was the initial conformation before optimization? If no why they state that they do so?
3. (section 2.1) The correct optimization procedure is one of the most important factors affecting quality of the obtained results. Its description in the manuscript is far from satisfactory! Compounds like PCL are expected to exist in conformational equilibrium of many conformers (many single bonds). It is essential to find the lowest energy conformers and consider them in computations of energy of formation. It is not stated also how initial configurations for DFT optimizations were generated. This is crucial for reliability of results. On line 75 authors state that local energy minima were confirmed. You should not look for local energy minima, but the global one and therefore efficient search for global energy minimum is crucial in such research. It might also be the case (here very probable) that compounds and complexes exist in complicated conformational equilibrium and it must be considered in evaluation of free energy of formation. This remark is related to the line 154-155 where authors state "...a conformer distribution and the selection of the most stable conformer were achieved." How?
Other comments:
4. Most of figures requires corrections to make them more informative. For example - Figure 1 - what is presented in the figure? Starting conformation for optimization or is it just a scheme? Figure 2 - is really beta-CDNH2 shown in Figure 2c??? What is the meaning of numbers shown in the figure? Figure3 - where is EGF in this figure? PCL is hardly visible - it should be marked with different color. What is the meaning of numbers shown in the figure? Figure 4 and 5 - authors compare calculated spectra to experimental. Why don't they put both of them in the same figure? It is the best and most informative method of comparison. also a-e letters are missing. Figure 8 and 9 - where is PCL in this figure? It is hard to see anything in this figure. c letter is missing in Fig 8. Fig 10 - what is the purpose of marking some regions of EPS? Are they important? If yes why? Again it is impossible to see where is PCL.
5. Units - authors use 4 different energy units - Hartree, eV, kcal/mol, kJ/mol. I understand that it is better to provide some values in Hartrees and some in eV, but why in some places there are kcal/mol and in others kJ/mol? Also electrostatic potential and dipole moment are not unitless physical quantities!
6. Why equations are in boldface type? Eqs 5-7 are all in row with numbers. I've never seen such a style in any publication. All symbols in equations should be carefully explained.
7. Line 153 - what is the purpose of repeating this information n-th time in the manuscript? I guess that by this time reader should be aware of using computational chemistry by authors.
8. Line 154 - "starting material molecules" sounds really strange. Usually substrates word is used instead.
9. Line 177 - again how initial structures for DFT optimization were prepared?
10. Line 180 - some coordinates in 3D space are provided here. These number are useless if coordinate system is not defined! How localization of PCL is defined? Do the authors mean - center of mass, center of geometry? It should be precisely defined.
11. Table 1 - Why dipole moments of complexes are lower than dipole moments of B-CDs? It can partially be a results of specific positioning of PCL in B-CD for which DM of PCL partially cancels DM of B-CD. But judging by number it is not the only effect. It should be investigated.
12. Lines 197-198 - Importantly here authors state that they achieved agreement between some computational results and experiment (with references to experiment). It is VERY IMPORTANT to state exactly what physical observables and to what extent MATCH between theory and experiment. Please provide observables with their values and differences!
13. Line 206 - what is contact angle?
14. Line 216 - "...values of interaction energy and consequently spontaneous formation." Yes, but in vacuum! Many complexes stable in gas phase are not stable in solution, because of differences in solvation energy.
15. Table 2 - Why values are not in rows? It is hard to judge which value corresponds to which compound. I guess that the value 1.06 of equilibrium constant is related to the value of deltaG = -14.777 kJ/mol. How was it calculated?
16. Lines 288-289 - The difference in free energy of formation between b-CNH2 and b-CD is huge considering very small difference in chemical composition between these compounds. What is the physical reason of such a huge difference?
17. Line 397 - units and exponents!
18. There is an error in ref 33. All references should be carefully checked.
Author Response
Reviewer: (line 23) What's the point of reporting absolute electronic energies of complexes in the abstract? What kind of information do these quantities carry? IMO these values (strongly method-dependent) calculated for three different compounds carry absolutely no valuable information.
Us: Done, we agree with the referee comment, therefore the electronic energies of complexes by interaction energies were changed.
Reviewer: Authors state that one of the investigated complexes is EGF-betaCDNH2/PCL (abstract and line 55). What exactly is EGF. Some minimal information about it should be provided. I assume that it is small protein. If this is the case - do authors really performed DFT calculations of complex with this small protein? If yes what was the initial conformation before optimization? If no why they state that they do so?
Us: We have already explained the EGF mean, explaining in addition the employed model for the study, please see the corrected manuscript.
Reviewer: (section 2.1) The correct optimization procedure is one of the most important factors affecting quality of the obtained results. Its description in the manuscript is far from satisfactory! Compounds like PCL are expected to exist in conformational equilibrium of many conformers (many single bonds). It is essential to find the lowest energy conformers and consider them in computations of energy of formation. It is not stated also how initial configurations for DFT optimizations were generated. This is crucial for reliability of results. On line 75 authors state that local energy minima were confirmed. You should not look for local energy minima, but the global one and therefore efficient search for global energy minimum is crucial in such research. It might also be the case (here very probable) that compounds and complexes exist in complicated conformational equilibrium, and it must be considered in evaluation of free energy of formation. This remark is related to the line 154-155 where authors state "...a conformer distribution and the selection of the most stable conformer were achieved." How?
Us: The methodology section has been improved. In addition, the redaction of the marked lines.
Other comments:
Reviewer: Most of figures requires corrections to make them more informative. For example - Figure 1 - what is presented in the figure? Starting conformation for optimization or is it just a scheme? Figure 2 - is really beta-CDNH2 shown in Figure 2c??? What is the meaning of numbers shown in the figure? Figure3 - where is EGF in this figure? PCL is hardly visible - it should be marked with different color. What is the meaning of numbers shown in the figure? Figure 4 and 5 - authors compare calculated spectra to experimental. Why don't they put both of them in the same figure? It is the best and most informative method of comparison. also a-e letters are missing. Figure 8 and 9 - where is PCL in this figure? It is hard to see anything in this figure. c letter is missing in Fig 8. Fig 10 - what is the purpose of marking some regions of EPS? Are they important? If yes why? Again it is impossible to see where is PCL.
Us: The information required for each figure has been already included:
- Figure 1- It is just a scheme- Line 71
- Figure 2 -done, line 198
- Figure 3-Done, see figure 3
- What is the meaning of numbers shown in the figure? Line 197 and 202
- authors compare calculated spectra to experimental. Why don't they put both in the same figure? It is the best and most informative method of comparison -since the experimental spectra were reported in previous works, we think not appropriate repeat this results in this paper.
- Figure 8 and 9 - where is PCL in this figure? Done, please see the new figures 8 and 9
- Fig 10 - what is the purpose of marking some regions of EPS? Are they important? Regarding to the EPS regions we are marking the regions with bigger and lower values such reactivity zones, and searching shows the differences among complexes.
Reviewer: Units - authors use 4 different energy units - Hartree, eV, kcal/mol, kJ/mol. I understand that it is better to provide some values in Hartrees and some in eV, but why in some places there are kcal/mol and in others kJ/mol? Also, electrostatic potential and dipole moment are not unitless physical quantities!
Us: The units-commented, have been adjusted, in Hartrees and kcal/mol, being convenient to explain that kcal/mol is used to describe energy in biological process and more common in diverse science areas. The units of dipole moment and electrostatic potential were also added.
Reviewer: Why equations are in boldface type? Eqs 5-7 are all in row with numbers. I've never seen such a style in any publication. All symbols in equations should be carefully explained.
Us: The format of the equations has been conveniently corrected; in addition the symbols were explained.
Reviewer: Line 153 - what is the purpose of repeating this information n-th time in the manuscript? I guess that by this time reader should be aware of using computational chemistry by authors.
Us: We conveniently attended the referee observation.
Reviewer: Line 154 - "starting material molecules" sounds really strange. Usually, substrates word is used instead.
Us: We sincerely agree with the commentary, please see the corrected manuscript.
Reviewer: Line 177 - again how initial structures for DFT optimization were prepared?
Us: The methodology was improved, in the shown line.
Reviewer: Line 180 - some coordinates in 3D space are provided here. These number are useless if coordinate system is not defined! How localization of PCL is defined? Do the authors mean - center of mass, center of geometry? It should be precisely defined.
Us: Done, it has been precisely defined the term, indicating that is the diameter of the cyclodextrin
Reviewer: Table 1 - Why dipole moments of complexes are lower than dipole moments of B-CDs? It can partially be a results of specific positioning of PCL in B-CD for which DM of PCL partially cancels DM of B-CD. But judging by number it is not the only effect. It should be investigated.
Us: We added a reference, explaining the dipole moment change. Kindly, see perspectives indicated in the conclusions.
Reviewer: Lines 197-198 - Importantly here authors state that they achieved agreement between some computational results and experiment (with references to experiment). It is VERY IMPORTANT to state exactly what physical observables and to what extent MATCH between theory and experiment. Please provide observables with their values and differences!
Us: Done, we improved the discussion. please see the corrected manuscript
Reviewer: Line 206 - what is contact angle?
Us: In materials science, the contact angle establishes the tangent (angle) of a liquid drop with a solid surface at the base, it is a measurement correlated with the wettability.
Reviewer: Line 216 - "...values of interaction energy and consequently spontaneous formation." Yes, but in vacuum! Many complexes stable in gas phase are not stable in solution, because of differences in solvation energy.
Us: We indicated in the manuscript, that the experimental results of a previously work of our research group, were performed in the solid nanofibers; in this sense, the calculation were achieved in solvent absence. However, considering the reviewer comment we have considered a future work bearing in mind a solvent effect.
Reviewer: Table 2 - Why values are not in rows? It is hard to judge which value corresponds to which compound. I guess that the value 1.06 of equilibrium constant is related to the value of deltaG = -14.777 kJ/mol. How was it calculated?
Us: Conveniently corrected the Table 2. The equilibrium constant was calculated by Gibbs free energy equation:
DG= -RTLnK
Reviewer: Lines 288-289 - The difference in free energy of formation between b-CNH2 and b-CD is huge considering very small difference in chemical composition between these compounds. What is the physical reason of such a huge difference?
Us: We think this minor difference is appropriate since the size (dimension) of the involved functional groups is minor (NH2 vs OH)
Reviewer: Line 397 - units and exponents!
Us: Done, please see the corrected manuscript
Reviewer: There is an error in ref 33. All references should be carefully checked.
Us: Done, please see the corrected manuscript.
Round 2
Reviewer 1 Report
This manuscript can be acceptted.
Reviewer 4 Report
The manuscript has been improved and can be published.